# SARS-CoV-2 Entry Genes Expression in Relation with Interferon Response in Cystic Fibrosis Patients

**DOI:** 10.3390/microorganisms9010093

**Published:** 2021-01-03

**Authors:** Camilla Bitossi, Federica Frasca, Agnese Viscido, Giuseppe Oliveto, Mirko Scordio, Laura Belloni, Giuseppe Cimino, Valeria Pietropaolo, Massimo Gentile, Gabriella d’Ettorre, Fabio Midulla, Maria Trancassini, Guido Antonelli, Alessandra Pierangeli, Carolina Scagnolari

**Affiliations:** 1Laboratory of Virology, Department of Molecular Medicine, Istituto Pasteur Italia, Sapienza University, 00161 Rome, Italy; camilla.bitossi@uniroma1.it (C.B.); federica.frasca@uniroma1.it (F.F.); agnese.viscido@uniroma1.it (A.V.); giuseppe.oliveto@uniroma1.it (G.O.); mirko.scordio@uniroma1.it (M.S.); massimo.gentile@uniroma1.it (M.G.); guido.antonelli@uniroma1.it (G.A.); alessandra.pierangeli@uniroma1.it (A.P.); 2Department of Internal Medicine, Sapienza University, 00161 Rome, Italy; laura.belloni@uniroma1.it; 3Lazio Reference Center for Cystic Fibrosis, Policlinico Umberto I University Hospital, Sapienza University, 00161 Rome, Italy; ciminolo@tiscali.it; 4Department of Public Health and Infectious Diseases, Sapienza University, 00161 Rome, Italy; valeria.pietropaolo@uniroma1.it (V.P.); gabriella.dettorre@uniroma1.it (G.d.); maria.trancassini@uniroma1.it (M.T.); 5Microbiology and Virology Unit, Policlinico Umberto I University Hospital, Sapienza University, 00161 Rome, Italy; 6Department of Maternal Science, Sapienza University of Rome, 00161 Rome, Italy; midulla@uniroma1.it

**Keywords:** ACE2, interferon, SARS-CoV-2, furin, TMPRSS2, cystic fibrosis, ISG15, dACE2

## Abstract

The expression rate of SARS-CoV-2 entry genes, angiotensin-converting enzyme 2 (ACE2), the main viral receptor and the proteases, furin and transmembrane serine protease 2 (TMPRSS2) in cystic fibrosis (CF) individuals is poorly known. Hence, we examined their levels in upper respiratory samples of CF patients (*n* = 46) and healthy controls (*n* = 45). Moreover, we sought to understand the interplay of type I interferon (IFN-I) with ACE2, furin and TMPRSS2 by evaluating their gene expression with respect to ISG15, a well-known marker of IFN activation, in upper respiratory samples and after ex vivo IFNβ exposure. Lower ACE2 levels and trends toward the reduction of furin and TMPRSS2 were found in CF patients compared with the healthy controls; decreased ACE2 amounts were also detected in CF individuals with pancreatic insufficiency and in those receiving inhaled antibiotics. Moreover, there was a strong positive correlation between ISG15 and ACE2 levels. However, after ex vivo IFNβ stimulation of nasopharyngeal cells, the truncated isoform (dACE2), recently demonstrated as the IFN stimulated one with respect to the full-length isoform (flACE2), slightly augmented in cells from CF patients whereas in those from healthy donors, dACE2 levels showed variable levels of upregulation. An altered expression of SARS-COV-2 entry genes and a poor responsiveness of dACE2 to IFN-I stimulation might be crucial in the diffusion of SARS-CoV-2 infection in CF.

## 1. Introduction

The impact of SARS-CoV-2 infection in CF patients remains not well characterized. Despite the fact that the CF condition could represent a risk of worse outcomes than the general population, an apparently low spread of SARS-CoV-2 has been reported in CF patients [1,2,3] as well as a wide range of clinical symptoms from completely asymptomatic to acute respiratory distress [2,3]. In this scenario, we previously found no case of SARS-CoV-2 infection in CF patients who attended clinic visits at the Regional (Lazio) CF Reference Center in Italy either with respiratory disease that required medical attention or in stable conditions, both before and after the Italian government restrictions for the first COVID-19 wave [4].

To infect respiratory cells, a SARS-CoV-2 Spike (S) protein engages the angiotensin-converting enzyme 2 (ACE2) and also employs two cellular proteases, furin and transmembrane serine protease 2 (TMPRSS2), for viral entry [5,6,7]. Notably, ACE2 expression has been shown to correlate with susceptibility to SARS-CoV-S-driven entry [8]. Moreover, several studies have indicated that interferons (IFNs) can upregulate the expression of both mRNA and protein levels of ACE2 [9,10,11]. More recently, a novel, transcriptionally independent truncated isoform of ACE2, which was designated as ΔACE2 (dACE2) but not the full-length ACE2 (flACE2), was demonstrated to be an IFN stimulated gene (ISG) [12]. However, cell type-specific mechanisms or kinetics can shape the different ability of type I IFN (IFN-I) to enhance ACE2 transcript, underlining the increasing complexity of this regulatory network [13].

In this study, we hypothesized that CF patients might exhibit a differential expression of ACE2, furin and TMPRSS2 in the upper respiratory tract cells, which might impact on the SARS-CoV-2 infection and/or spread. Moreover, as there are contrasting results on the relationship between IFN-I and ACE2 isoforms expression [13,14], we aimed to study the relationship of ACE2 (all isoforms), furin and TMPRSS2 with IFN stimulated gene 15 (ISG15), a well-established ISG that acts as a potential modifier of CF severity, as well as ACE2 responsiveness to ex vivo IFN-I stimulation [15]. 

## 2. Materials and Methods

### 2.1. Patients and Study Design 

Patients with a clinical diagnosis of CF attending the Regional (Lazio) CF Reference Center of the University Hospital Policlinico Umberto I were enrolled during the 2019/2020 winter season. Real-time (RT) PCR reactions targeting the RNA-dependent RNA polymerase (RdRp) and the envelope (E) genes of SARS-CoV-2 were developed in-house following published protocols [4,16]. Cells were collected from oropharyngeal swabs taken for routine diagnostic testing. Oropharyngeal swabs from sex and age matched healthy controls were also analyzed. 

### 2.2. Gene Expression Analysis 

The mRNA copy number of ACE2 (Hs.PT.58.27645939), flACE2 (F5’-GGATATGCCCCATCT CATGATG-3’, R5’-GGGCGACTTCAGGATCCTTAT-3’; Probe 5’-[56FAM]ATGGACGAC[ZEN] TTCCTGACAG[3IABkFQ]-3’), dACE2 (F5’-AGCTGTCAGGAAGTCGTCCATT-3’, R5’-GGAA GCAGGCTGGGACAAA-3’; Probe 5’-[56FAM]AGGGAGGAT[ZEN]CCTTATGTG[3IABkFQ]-3’), furin (Hs.PT.58.1294962), TMPRSS2 (Hs.PT.58.4661363) and ISG15 (F5’-TGGCGGGCAACG AATT-3’, R5’-TGATCTGCGCCTTCA-3’; Probe 5’-[6FAM]TGAGCAGCTCCATGTC[TAM]n-3) was measured by quantitative RT PCR assays. Primers and a probe for ACE2 (Hs.PT.58.27645939) bind internally (exons 14–15) to the mRNA sequence and can detect both truncated and full-length ACE2 isoforms. In patients’ diagnostic samples, because of the limited quantity, we analyzed the ACE2 expression (all isoforms) using the pre-designed assay, which did not discriminate between the two isoforms; by contrast, in cells obtained for ex vivo experiments, we also analyzed dACE2 and flACE2 levels to better understand the IFN-I ability to induce ACE2. The housekeeping gene β-glucuronidase/GUS was used as an internal control [17]. Transcript levels of ACE2 (all isoforms), furin, TMPRSS2 and ISG15 were calculated using the formula 2^-ΔCt^.

### 2.3. Ex Vivo IFN Induction Experiments

The experiments of IFN-I exogenous administration in cells from oropharyngeal swabs were performed treating 2 × 10^6^ cells/well in 48 well plates with IFNβ1a (Avonex Pen, Biogen Idec Ltd. Cambridge, Massachusetts, USA) at 5 ng/mL. The comparative threshold cycle method was applied to calculate the fold changes of ACE2 (all isoforms), flACE2, dACE2 and ISG15 compared with non-stimulated cells (2^-ΔΔCT^). All experiments were conducted in triplicate. 

### 2.4. Statistical Analysis 

The differences in gene expression levels between CF patients and controls and among CF groups were evaluated by a Mann–Whitney test. Spearman’s rho coefficient was calculated to assess correlations between the gene expression and patients’ data. A Wilcoxon signed-rank test was used to compare gene levels after ex vivo IFN stimulation. Analysis was performed with SPSSv.20.0 for Windows.

## 3. Results

A total of 46 CF patients and 45 healthy controls were evaluated for ACE2 (all isoforms), furin and TMPRSS2 gene expression; demographic and clinical characteristics are reported in Table 1. No SARS-CoV-2 positive result was identified in all of the respiratory samples analyzed. Lower ACE2 levels were found in CF patients compared with those in the control group (*p* = 0.034); a decreased but not statistically significant expression of furin and TMPRSS2 was observed in CF individuals (Table 1). In both the CF and healthy groups, a positive correlation between ACE2 and TMPRSS2 levels (r = 0.445, *p* = 0.002; r = 0.575, *p* < 0.001) was detected. ACE2 levels correlated with furin in CF patients (r = 0.581, *p* < 0.001) but not in healthy donors. 

The main CF clinical features were then evaluated in relation to the study genes. The levels of ACE2 expression did not differ between CF patients with or without *Pseudomonas aeruginosa* colonization (*p* = 0.414, Table 2). CF patients with pancreatic insufficiency had lower levels of ACE2 expression with respect to those not suffering from this CF-related metabolic deficiency (*p* = 0.037). Moreover, those receiving inhaled antibiotics, compared with patients who were not, displayed decreased levels of ACE2 and furin mRNAs (*p* = 0.006; *p* = 0.018). No other differences were found in the SARS-CoV-2 entry genes according to the clinical parameters of CF individuals (Table 2). 

As there is growing evidence suggesting the involvement of IFN-I pathways in CF lung dysfunctions [15] and given the complex relationship between IFN and the different ACE2 isoform expressions [9,12,13], ISG15 gene expression was assessed as a consolidated marker of IFN’s activation. Increased ISG15 levels were found in CF patients compared with healthy controls (*p* < 0.001, Table 1). A strong positive correlation was found between ISG15-mRNA levels and ACE2 isoforms in both CF patients (r = 0.473, *p* = 0.001) and healthy controls (0.336, *p* = 0.024). 

To further explore the relationship between ACE2 and the IFN-I response, we examined the ability of IFNβ to promote ex vivo the expression of ACE2 (all isoforms) and ISG15 in cells from oropharyngeal swabs of six CF subjects (median age: 33 years; 2 males) and six matched healthy controls (median age: 31.5 years; 3 males). Ex vivo IFNβ stimulation induced a median increase in ISG15 levels of 3- and 18-fold in CF patients and healthy controls, respectively (CF individuals: T0 vs T1 *p* = 0.030; healthy controls: T0 vs T1 *p* = 0.027, Figure 1B). By contrast, ACE2 augmented but not at significant levels (T0 vs T1 *p* = 0.116) in CF individuals, whereas ACE2 levels were increased by a median of 4-fold (T0 vs T1 *p* = 0.028) in the healthy individuals (Figure 1A). 

In order to discriminate which of the ACE2 isoforms could be upregulated by IFN-I, we then examined the ex vivo IFN-I mediated activation of dACE2 and flACE2 isoforms. As the oropharyngeal swabs used were residual samples derived from routine microbiological diagnostic testing and given that we did not have enough RNA extracted from the cultured cells, this analysis was performed in three out of six healthy donors and CF patients. Following the IFNβ treatments, variable levels of expression and upregulation of the dACE2 levels were observed in the cells collected from oropharyngeal swabs of healthy donors (Figure 2). On the other hand, the IFN-I mediated increase of the dACE2 isoform was moderate in all of the CF patients analyzed (Figure 2). Low levels of flACE2 were induced after IFNβ treatment in both CF patients and healthy controls (Figure 2).

## 4. Discussion

To the best of our knowledge, this is the first study to show reduced levels of ACE2 in the oropharyngeal cells of CF patients compared with healthy individuals; furin and TMPRSS2 also showed a trend toward reduction in CF patients. Intriguingly, host serine proteases participate in the regulation of the epithelial sodium channel (ENaC) in human airways cells [18]. In this regard, SARS-CoV-2 has been reported to mimic the proteolytic activation of EnaC [19], highlighting the complexity of the analyzed phenomenon. In this scenario, both ACE2 isoforms and furin levels decreased in CF individuals receiving inhaled antibiotics suggesting that antimicrobial agents might confer a benefit against COVID-19 [20]. Moreover, reduced levels of ACE2 (all isoforms) were shown to be associated with acute respiratory distress syndrome [21,22] and with decreased insulin secretion in diabetes progression [23], which is in agreement with our results in CF patients with pancreatic insufficiency.

We also found increased ISG15 levels in CF patients consistent with the higher rate of respiratory pathogens colonizing CF patients [15]. As SARS-CoV-2 is highly sensitive to the antiviral action of IFN-I [24], the upregulation of the IFN response in the airway tracts of CF patients might be involved in mediating protection against SARS-CoV-2 infection.

Overall, our data confirmed that a direct relationship between ACE2 and IFN-I pathways does exist in CF patients. As we could not discriminate the induction of each ACE2 isoform individually in the analysis performed in vivo, we measured the levels of ACE2 (all isoforms), flACE2, dACE2 and ISG15 after ex vivo stimulation with IFNβ in cells collected from oropharyngeal swabs of CF individuals and healthy donors. Of note, following the ex vivo stimulation of oropharyngeal cells with IFN-I, ACE2 (all isoforms) was induced at lower levels compared with ISG15. It is known that many cell types respond to IFN-I with varying transcriptional responses [25]. Thus, the ability of IFN-I to increase the expression of different ISGs including ACE2 can vary widely depending on the different cell type composition of the respiratory sample analyzed [13,14]. Furthermore, our data confirmed that flACE2 did not increase after IFNβ treatment [12]. In addition, CF patients exhibited a moderate IFN-I mediated induction of dACE2 while the healthy donors had a highly variable upregulation of dACE2 after IFNβ stimulation. Therefore, independent of the variability observed in the ex vivo upregulation of dACE2 and ISG15 after IFNβ stimulation, the responsiveness to IFN-I appeared to be slightly reduced in cells of the upper respiratory tract of CF patients compared with the healthy donors. It is possible that a hyper-stimulated status of the IFN pathways in the CF respiratory tract due to polymicrobial colonization might limit their capability to further mount an IFN response and also the increase of dACE2 levels.

In conclusion, lower ACE2 levels in CF upper respiratory cells might be one of the factors explaining the apparently lower incidence of SARS-CoV-2 infection in CF patients compared with the general population [3,4]. In apparent contrast, a recent study reported a remarkable upregulation of ACE2 (and TMPRSS2) expression in biopsies from CF lungs with respect to controls [11]. However, these measurements were in biopsies from CF patients undergoing lung transplantation that usually present a more severe respiratory disease, whereas this study’s samples were from oropharyngeal swabs taken for routine diagnostic testing. As recent observations highlighted the complexity of ACE2 regulation together with the existence of an IFN induced truncated isoform of this receptor [12,13], our results underline the need to acquire further data in CF patients. If confirmed, the reduced IFN-I capability to induce dACE2 in CF may support the importance of a well-balanced IFN response between the beneficial and detrimental effects for the possible risk of severe pulmonary disease.

## Figures and Tables

**Figure 1 microorganisms-09-00093-f001:**
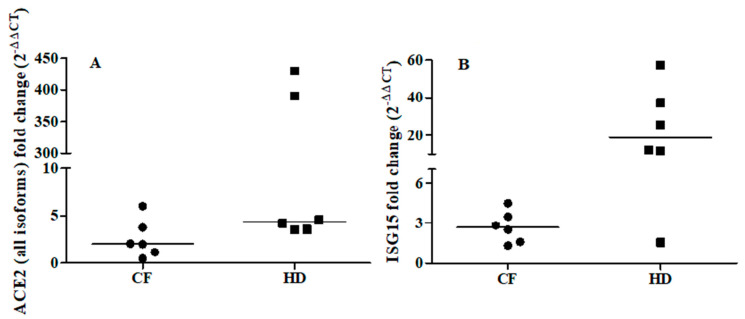
Fold change of ACE2 (all isoforms, Panel A) and ISG15 (Panel B) after ex vivo IFNβ stimulation of the oropharyngeal cells of cystic fibrosis (CF) patients (*n* = 6) and matched healthy controls (HD) patients (*n* = 6). ACE2 and ISG15 expressions were examined before (T0) and after (T1) IFNβ stimulation (5 ng/mL) for 24 h. ACE2: CF individuals, T0 vs T1 *p* = 0.116; HD individuals, T0 vs T1 *p* = 0.028 by Wilcoxon signed-rank test. ISG15: CF individuals, T0 vs T1 *p* = 0.030; HD individuals, T0 vs T1 *p* = 0.027 by Wilcoxon signed-rank test.

**Figure 2 microorganisms-09-00093-f002:**
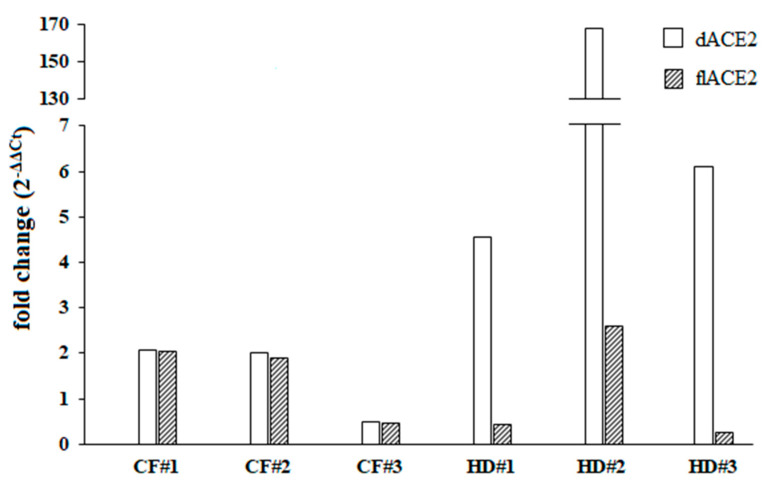
Fold change of the truncated angiotensin-converting enzyme 2 isoform (dACE2) and the full-length angiotensin-converting enzyme 2 isoform (flACE2) was shown individually for three different cystic fibrosis (CF) patients and healthy controls (HD) patients. Both dACE2 and flACE2 expression levels were examined before (T0) and after (T1) IFNβ stimulation (5 ng/mL) for 24 h.

**Table 1 microorganisms-09-00093-t001:** Demographic, genetic, clinical features and expression levels of angiotensin-converting enzyme 2 (ACE2), furin, transmembrane serine protease 2 (TMPRSS2) and Interferon (IFN) stimulated gene 15 (ISG15) of cystic fibrosis (CF) patients (*n* = 46) and healthy controls (*n* = 45).

Items	Cystic Fibrosis *n* = 46	Healthy Controls *n* = 45
Age (years)	35.04 ± 12.13	44.82 ± 15.00
Male (%)	24/46 (52.17 %)	23/45 (51.11%)
ΔF508 homozygous/ΔF508 heterozygous/others, n	19/16/11	N.A.
Pancreatic insufficiency, n (%)	30 (65.21%)	N.A.
BMI, mean (SD)	22.01 (3.38%)	N.A.
FEV1% ≥ 70%, n (%)	28 (60.87%)	N.A.
40% > FEV1% < 70%, n (%)	12 (26.09%)	N.A.
FEV1% ≤ 40%, n (%)	6 (13.04%)	N.A.
Hospitalization, n (%)	3 (6.52%)	N.A.
Inhaled antibiotics, n (%)	17 (36.95%)	N.A.
Oral antibiotics, n (%)	11 (23.91%)	N.A.
IV antibiotics, n (%)	2 (4.34%)	N.A.
*P. aeruginosa* colonization, n (%)	21 (45.65%)	N.A.
**Gene Expression**		
ACE2	0.006 (0.001–0.024)	0.015 (0.003–0.05)◊
Furin	0.003 (0.0006–0.01)	0.004 (0.0007–0.01)
TMPRSS2	0.119 (0.01–0.62)	0.241 (0.02–0.61)
ISG15	9.986 (3.03–32.79)□	2.321 (0.93–4.60)

Data are presented as a number or mean. N.A.: not applicable. FEV1%: Forced Expiratory Volume. According to the European Respiratory Society criteria: FEV1 ≥ 70% mild obstruction or normal, 40% > FEV1% < 70% moderate obstruction and FEV1% ≤ 40% severe obstruction. Transcript levels of ACE2, TMPRSS2, furin and ISG15 were calculated using the formula 2^-ΔCt^ and indicated as a median (interquartile range = IQR). ◊Healthy controls displayed higher levels of ACE2 mRNA (*p* = 0.034 by Mann–Whitney U test). □CF patients had higher levels of ISG15 expression (*p* < 0.001 by Mann–Whitney U test).

**Table 2 microorganisms-09-00093-t002:** Expression levels of ACE2, furin, TMPRSS2 and ISG15 in cystic fibrosis (CF) patients (*n* = 46), according to the main clinical conditions.

Clinical Condition	ACE2	Furin	TMPRSS2	ISG15
No pancreatic insufficiency	0.015 (0.005–0.025)	0.004 (0.002–0.023)	0.105 (0.026–0.324)	10.32 (3.323–17.230)
Pancreatic insufficiency	0.003 (0.0008–0.016)*	0.002 (0.0001–0.015)	0.153 (0.008–0.837)	10.68 (2.050–44.332)
FEV1% ≥ 70%	0.006 (0.002–0.023)	0.003 (0.001–0.017)	0.118 (0.017–0.389)	11.392 (2.789–33.358)
40% > FEV1% < 70%	0.011 (0.001–0.022)	0.001 (0.0004–0.015)	0.301 (0.027–0.964)	16.393 (7.775–32.219)
FEV1% ≤ 40%	0.011 (0.001–0.032)	0.004 (0.001–0.014)	0.027 (0.012–0.521)	4.704 (0.518–43.325)
No inhaled antibiotics	0.009 (0.004–0.026)	0.004 (0.002–0.017)	0.183 (0.070–0.705)	14.370 (4.208–41.224)
Inhaled antibiotics	0.0009 (0.0005–0.004)●	0.0008 (0.0002–0.002)●	0.013 (0.005–0.389)	2.602 (0.952–16.223)
No oral antibiotics	0.004 (0.001–0.015)	0.002 (0.0003–0.006)	0.118 (0.011–0.449)	7.960 (3.228–28.307)
Oral antibiotics	0.004 (0.001–0.027)	0.004 (0.0005–0.013)	0.100 (0.012–0.728)	12.041 (1.409–37.600)
*P. aeruginosa* non-colonized	0.007 (0.002–0.018)	0.004 (0.001–0.017)	0.178 (0.034–0.683)	14.420 (4.287–39.670)
*P. aeruginosa* colonized	0.004 (0.0008–0.030)	0.001 (0.0004–0.025)	0.027 (0.005–0.618)	7.464 (1.536–19.234)

Transcript levels of ACE2, furin, TMPRSS2 and ISG15 were calculated using 2^-ΔCt^ and indicated as median (interquartile range = IQR). *CF patients with pancreatic insufficiency had lower levels of ACE2 expression (*p* = 0.037 by Mann–Whitney U test). ●CF patients receiving inhaled antibiotics displayed lower levels of ACE2 and furin mRNAs (*p* = 0.006; *p* = 0.018 by Mann–Whitney U test).

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
