# Peer review of "SARS-CoV-2 Entry Genes Expression in Relation with Interferon Response in Cystic Fibrosis Patients"

_microorganisms, 2021, doi:10.3390/microorganisms9010093_

Round 1

Reviewer 1 Report

The manuscript ID microorganisms-1055251 reports the expression rate of SARS-CoV-2 entry genes, ACE2in cystic fibrosis (CF) individuals. The document is well written and captured the most recent pieces of literature which is excellent. Hence convey my congratulator message to Bitossi et al. for the excellent work done. However, I have a concern with regards to the sample size n=46? Why 46??

Also, the manuscript needs minor grammar checks

Minor corrections

Line 22 include “by” before evaluating

Line 47 please correct the symbol of the delta

Line 64 please include the approval number

Line 81 please check 2x106 ……6 is supposed to be superscript

  • Also, I suggest the authors cite “The Structural Basis of Accelerated Host Cell Entry by SARS‐CoV‐2 (DOI:1111/FEBS.15651)

Author Response

Reviewer # 1: The manuscript ID microorganisms-1055251 reports the expression rate of SARS-CoV-2 entry genes, ACE2in cystic fibrosis (CF) individuals. The document is well written and captured the most recent pieces of literature which is excellent. Hence convey my congratulator message to Bitossi et al. for the excellent work done. However, I have a concern with regards to the sample size n=46? Why 46?? Also, the manuscript needs minor grammar checks.

Response: We thank the Reviewer#1 for his comments. English has been controlled and adjusted. In relation to the sample size, since all test samples are residuals collected during microbiological diagnostic routine, only a limited number of specimens reached sufficient volume necessary for SARS-CoV-2 investigation and gene expression analysis.   

- Line 22 include “by” before evaluating

Response: We thank Reviewer#1, the sentence has been corrected (page 1, line 22). 

- Line 47 please correct the symbol of the delta

Response: We thank Reviewer#1, the symbol has been added to the text (page 2, line 47). 

- Line 64 please include the approval number

Response: We thank Reviewer#1 for this comment and we apologize for this oversight. The committee approval number has been added to the text (page 2, 64).

- Line 81 please check 2x106 ……6 is supposed to be superscript

Response: We thank Reviewer#1, this has been corrected in the text (page 2, line 81).

 - Also, I suggest the authors cite “The Structural Basis of Accelerated Host Cell Entry by SARSCoV2 (DOI:1111/FEBS.15651)

Response: We thank the Reviewer#1 for this suggestion. The reference has been added to the text on page 1, line 45.

Reviewer 2 Report

It has been thought that patients with CF condition would be at high risk if they were infected with CoV-SARS-2. In this study, authors evaluated the gene expression levels of the major components related to CoV-SAR-2 infection between the CF patients and health individuals. They found a relatively lower level of ACE2 and trends toward reduction of furin and TMPRSS2 in the CF group, as well as demonstrated a relationship between ACE2 and IFN-1 response. The authors previously found a scenario that no case of infection was reported in the CF patients who attended clinical visits at the Regional CF Reference Center in Italy. It could be too early to conclude that lower level of ACE2 expression in upper respiratory cells could be a factor related to the low incidence of SAR2 infection in CF patients, this short communication is an interesting study and provides valuable data to the CF research community. Reviewer recommends its publication in Microorganisms.

Minor: line 81, 2x10"6" should be superscripted

Author Response

Reviewer # 2: It has been thought that patients with CF condition would be at high risk if they were infected with CoV-SARS-2. In this study, authors evaluated the gene expression levels of the major components related to CoV-SAR-2 infection between the CF patients and health individuals. They found a relatively lower level of ACE2 and trends toward reduction of furin and TMPRSS2 in the CF group, as well as demonstrated a relationship between ACE2 and IFN-1 response. The authors previously found a scenario that no case of infection was reported in the CF patients who attended clinical visits at the Regional CF Reference Center in Italy. It could be too early to conclude that lower level of ACE2 expression in upper respiratory cells could be a factor related to the low incidence of SAR2 infection in CF patients, this short communication is an interesting study and provides valuable data to the CF research community. Reviewer recommends its publication in Microorganisms.

 Response: We thank Reviewer#2 for his/her positive comments.  

- line 81, 2x10"6" should be superscripted

Response: We thank Reviewer#2 for this comment, this has been corrected in the text (page 2, line 81).